# Climate Change Increases the Expansion Risk of *Helicoverpa zea* in China According to Potential Geographical Distribution Estimation

**DOI:** 10.3390/insects13010079

**Published:** 2022-01-11

**Authors:** Haoxiang Zhao, Xiaoqing Xian, Zihua Zhao, Guifen Zhang, Wanxue Liu, Fanghao Wan

**Affiliations:** 1State Key Laboratory for Biology of Plant Diseases and Insect Pests, Institute of Plant Protection, Chinese Academy of Agricultural Science, Beijing 100193, China; zhao834323482@163.com (H.Z.); xianxiaoqing1981@163.com (X.X.); zhangguifen@caas.cn (G.Z.); wanfanghao@caas.cn (F.W.); 2Department of Entomology, College of Plant Protection, China Agricultural University, Beijing 100193, China; zhzhao@cau.edu.cn

**Keywords:** *Helicoverpa zea*, climate change, suitable habitat, MaxEnt model, calibration

## Abstract

**Simple Summary:**

*Helicoverpa* *zea* is one of the most destructive lepidopteran agricultural pests in the world and can disperse long distances both with and without human transportation. It is listed in the catalog of quarantine pests for plants imported to the People’s Republic of China but has not yet been reported in China. On the basis of 1781 global distribution records of *H. zea* and eight bioclimatic variables, we predicted the potential geographical distributions (PGDs) of *H. zea* by using a calibrated MaxEnt model. The results showed that the PGDs of *H. zea* under the current climate are large in China. Future climate changes under shared socioeconomic pathways (SSP) 1-2.6, SSP2-4.5, and SSP5-8.5 for both the 2030s and 2050s will facilitate the expansion of PGDs for *H. zea*. *Helicoverpa zea* has a high capacity for colonization by introduced individuals in China. Customs ports should pay attention to the host plants of *H. zea* and containers harboring this pest.

**Abstract:**

*Helicoverpa zea*, a well-documented and endemic pest throughout most of the Americas, affecting more than 100 species of host plants. It is a quarantine pest according to the Asia and Pacific Plant Protection Commission (APPPC) and the catalog of quarantine pests for plants imported to the People’s Republic of China. Based on 1781 global distribution records of *H. zea* and eight bioclimatic variables, the potential geographical distributions (PGDs) of *H. zea* were predicted by using a calibrated MaxEnt model. The contribution rate of bioclimatic variables and the jackknife method were integrated to assess the significant variables governing the PGDs. The response curves of bioclimatic variables were quantitatively determined to predict the PGDs of *H. zea* under climate change. The results showed that: (1) four out of the eight variables contributed the most to the model performance, namely, mean diurnal range (bio2), precipitation seasonality (bio15), precipitation of the driest quarter (bio17) and precipitation of the warmest quarter (bio18); (2) PGDs of *H. zea* under the current climate covered 418.15 × 10^4^ km^2^, and were large in China; and (3) future climate change will facilitate the expansion of PGDs for *H. zea* under shared socioeconomic pathways (SSP) 1-2.6, SSP2-4.5, and SSP5-8.5 in both the 2030s and 2050s. The conversion of unsuitable to low suitability habitat and moderately to high suitability habitat increased by 8.43% and 2.35%, respectively. From the present day to the 2030s, under SSP1-2.6, SSP2-4.5 and SSP5-8.5, the centroid of the suitable habitats of *H. zea* showed a general tendency to move eastward; from 2030s to the 2050s, under SSP1-2.6 and SSP5-8.5, it moved southward, and it moved slightly northward under SSP2-4.5. According to bioclimatic conditions, *H. zea* has a high capacity for colonization by introduced individuals in China. Customs ports should pay attention to host plants and containers of *H. zea* and should exchange information to strengthen plant quarantine and pest monitoring, thus enhancing target management.

## 1. Introduction

With the increasing number of non-native species and global trade integration development, global biological invasions are considered to be one of the important factors contributing to the decline of biodiversity and loss of ecosystem functions [1,2,3]. Biological invasions are often influenced by various factors of global change, especially climate warming [4,5,6]. With climate warming, invasive alien species (IAS) become more environmentally adapted and more competitive with native species after successful establishment [7,8]. Climate warming increases the survival of IAS and the availability of ecological niches in the invaded regions, which promotes the invasive probability of IAS [9,10,11]. In addition, there are differences in phenology between invasive and native species [12]. Changes in precipitation and temperature patterns due to climate warming allow IAS to adapt more quickly to invaded environments and enhance their ability to disperse [13,14,15]. According to reports provided by the Sixth Assessment Report of the Intergovernmental Panel on Climate Change (IPCC), warming will reach or exceed 1.5 °C in the next few decades [16]. As climate warming intensifies, biological invasions will become more rapid, more intense and more harmful. There is not abundant proof that the number of IAS will decrease [17]. Therefore, predicting the potential geographic distributions (PGDs) of IAS and assessing their potential for future invasion and dispersal are key problems worth solving.

Species distribution models (SDMs) have been widely used to study autecology and the effects of climate change on PGDs [18,19]. SDMs are also playing an increasingly important role in predicting the PGDs of species under climate change [20,21]. The MaxEnt model uses species distribution records and the corresponding environmental variables in a given habitat and is very suitable for modeling the PGDs of species [22,23]. In recent years, application of the model has expanded not only to examine ecological degradation processes, such as biological invasions [24] and ecological damage [25], but also to predict potential risk areas for pests and epidemic disease [26,27]. Research on the identification of habitats at risk for biological invasion has gradually become a popular research topic [28]. In previous studies identifying risk areas for IAS, researchers inferred the PGDs of IAS and combined the results with ArcGIS software to identify locations with a high risk of invasion [29].

*Helicoverpa* mainly consists of *H. zea*, *H. virescens*, *H. armigera*, *H. assulata*, *H. viriplaca* and *H. punctigera*. Among these six species, *H. armigera*, *H. assulata* and *H. viriplaca* have been recorded in China, and they cause great economic losses of the agricultural crops annually [30,31]. At present, the distribution of *H. zea* has not been reported in China. China and North America have similar climates, and most species from North America can quickly adapt to new habitats in China and successfully colonize in a relatively short period after being introduced [32]. *Helicoverpa*
*zea*, a well-documented and endemic pest throughout most of South and North America, and can affect 100 species of host plants, including important agricultural crops, such as corn, cotton, and wheat [33]. It has not only been listed as a quarantine pest of Asia and Pacific Plant Protection Commission (APPPC, 2021), but was also recently added to the European and Mediterranean Plant Protection Organization A1 list of quarantine pests [34]. China customs listed *H. zea* in the catalog of quarantine pests for plants imported to the People’s Republic of China in 2006. Due to its high reproductive capacity, wide range of host plants, and seasonal migration of adults, *H. zea* has the potential to greatly harm Chinese economic crops if introduced into the country.

To date, studies on *H. zea* have mainly focused on its morphological features [35] and biological characteristics [33], while few studies have aimed to identify its invasive risk in different areas. In this study, distribution data of *H. zea* and related environmental data were used to identify the PGDs of *H. zea* in China based on the MaxEnt model and ArcGIS software, and we aimed to investigate the following issues: (1) the relationship between the PGDs of *H. zea* and environmental variables; (2) the PGDs of *H. zea* in China under the current climate; and (3) changes in the PGDs of *H. zea* in China under climate change, and the shifting trend of PGDs of *H. zea*. Based on the above results, the dynamic characteristics and the significant environmental variables limiting the PGDs of *H. zea* in China under climate change were clarified, and the possibility of its dispersal in China was assessed. Our study provided a scientific and theoretical basis for establishing early warning monitoring of *H. zea* in China.

## 2. Materials and Methods

### 2.1. Distribution Records of H. zea

Distribution records of *H. zea* were first collected from the Global Biodiversity Information Facility (GBIF, https//:www.gbif.org/, accessed on 5 November 2021) and the Invasive Species Compendium of the Center for Agriculture and Bioscience International (CABI-ISC, https//:www.cabi.org/isc, accessed on 5 November 2021). There were a total of 5899 distribution records for *H. zea*. Duplicate records and distribution points without detailed geographic locations were removed from the dataset. ENMTools software (http://purl.oclc.org/enmtools, accessed on 7 November 2021) was used to select distribution records of *H. zea* for model simulation. With reference to the resolution of the environmental variables, only one distribution point was retained within each 5 km × 5 km raster. Finally, 1781 valid occurrence records of *H. zea* were retained to prevent model overfitting (Figure 1).

### 2.2. Environmental Variables, Map and Model

Nineteen bioclimatic variables and altitude variables were downloaded from the World Climate Database (version 2.1, http://www.worldclim.org//, accessed on 5 November 2021) with a resolution of 2.5′ (Table 1). This database collected detailed meteorological information from meteorological stations around the world from 1970-2000. Future climate data were obtained using the BCC-CSM2-MR global climate model developed by the National Climate Center for two periods (2030s and 2050s) and three shared socioeconomic pathways (SSP1-2.6, SSP2-4.5, and SSP5-8.5). The shared socioeconomic pathways suggest, for example, that a future with “resurgent nationalism” and fragmentation of the international order could make the “well below 2°C” Paris target impossible [16]. An explanation of the shared socioeconomic pathways is provided in Table 2. The world administrative map was downloaded from the National Earth System Science Data Center, National Science and Technology Infrastructure of China (http://www.geodata.cn, accessed on 5 November 2021), and MaxEnt 3.4.4, which is freely available online (http://biodiversityinformatics.amnh.org/open_source/MaxEnt/, accessed on 5 November 2021).

Because there were some linear correlations between climate variables, correlation analysis of the 19 bioclimatic variables was performed by ENMTools software [36,37]. Selection of bioclimatic variables was carried out in two steps: (1) the bioclimatic variables and altitude variable were imported into the MaxEnt model three times, and those with zero contribution were removed; and (2) all bioclimatic variables and altitude variable with contribution rates greater than 0 were selected for correlation analysis in ENMTools. When the correlation coefficient of two bioclimatic variables was greater than or equal to 0.8, the variable with the highest score was retained; eight variables were ultimately retained for MaxEnt modeling (Table 1, Figure 2).

### 2.3. MaxEnt Model Calibration

MaxEnt, also known as the maximum entropy model, is an ecological niche model based on the theory of maximum entropy and constructed on the Java platform [38]. The most important parameters of MaxEnt are feature classes (FCs) and the regularization multiplier (RM). The calibration of FCs and the RM can significantly improve the prediction accuracy of the MaxEnt model [39,40]. In our work, the MaxEnt model was calibrated by setting different combinations of FCs and incremental RMs. The FCs included five basic parameters: linear-L, quadratic-Q, product-P, threshold-T, and hinge-H. There were 31 different combinations. Briefly, the RM is set to 4 or less and uses an interval of 0.1, increasing from 0.1 to 4, for a total of 40 values in this paper. The Kuenm package of R software (https://www.r-project.org/, accessed on 5 November 2021) was used to create the 1240 candidate models [41]. Finally, R software was used to select the significant models with omission rates less than 5% and delta AICc values less than 2.

### 2.4. Model Settings and Evaluation

An optimal model was obtained after MaxEnt model calibration. For this, 25% of the distributed points were used to test the MaxEnt model, and the remaining 75% were used to train the MaxEnt model [40]. The maximum number of iterations was 500, and the maximum number of background points was 10,000 in the MaxEnt model. The importance of the environmental variables limiting H. zea was assessed by the contribution rates and the jackknife method from the MaxEnt model. Receiver Operating Characteristic (ROC) curves and Area Under the ROC Curve (AUC) values were used to test the accuracy of the model output. The ROC curve is an acceptance curve with the horizontal coordinate indicating the false positive rate (1-specificity) and the vertical coordinate indicating the true positive rate (1-omission rate) [42]. The AUC values not affected by thresholds are more objective than others for model assessment. An AUC value closer to 1 indicates that the model result is better. The evaluation criteria of model simulation accuracy were as follows: poor (AUC ≤ 0.50), available (0.5 < AUC ≤ 0.80) and excellent (0.80 < AUC ≤ 1.00) [43].

Among the results of MaxEnt modeling, the maximum value of 10 replicates was selected as the final result in this study. The ASCII raster layer was generated based on the value of the presence probability (*p*) of H. zea, which ranged from 0 to 1. A higher presence probability of H. zea was indicated by a higher P value. The results were converted into raster format in ArcGIS software (https://www.arcgis.com, accessed on 5 November 2021) and extracted according to the administrative division map of China. Finally, the suitable habitats were ranked and visualized. The suitable areas were classified into four types: high suitability habitat (0.5 < *p* ≤ 1.0), moderate suitability habitat (0.5 ≤ *p* < 0.3), low suitability habitat (0.3 ≤ *p* < 0.1) and unsuitable habitat (0.0 < *p* ≤ 0.1). The number of grids for each type and the proportion of suitable habitats in each class were calculated.

## 3. Results

### 3.1. FC and RM of the Optimal Model

The results of R software analysis showed that 1160 of the 1240 selected candidate models were statistically significant, and the optimal model with the smallest delta AICc value was selected. The FC was L, Q, and RM was 0.3 in the optimal models (Figure 3). On the basis of 1781 distribution records of *H. zea*, the suitable habitats for *H. zea* were simulated using the MaxEnt model under the current climate and projected climate change. The results showed that all of the mean AUC values of the MaxEnt model were approximately 0.9 (Appendix A), indicating that the model fit was excellent.

### 3.2. Significant Environmental Variables

The percentage contribution of variables to the model fit and regularized training gain were combined to identify significant environmental variables. The top three variables with the highest percent contribution were precipitation seasonality (Bio15, 37.8%), mean diurnal range (Bio2, 24.9%) and precipitation of the driest quarter (Bio17, 13.6%), with a cumulative contribution of 76.3% (Appendix A). The results of the jackknife method revealed that the three most significant effects on regularized training gain with only one variable were precipitation of the driest quarter (Bio17), precipitation of the warmest quarter (Bio18), and precipitation seasonality (Bio15), indicating that these variables provided information that the other variables did not (Figure 4). The significant environmental variables affecting the potential suitable habitats were one temperature (mean diurnal range) and three precipitation variables (precipitation seasonality, precipitation of the driest quarter, and precipitation of the warmest quarter).

The relationships between the presence probability of *H. zea* and environmental variables were determined on the basis of the response curves of environmental variables to presence probability (Figure 5). When the presence probability of *H. zea* was greater than the threshold for high suitability habitat classification (*p* ≥ 0.5), the corresponding interval was suitable for the survival and growth of *H. zea*. The mean diurnal range suitable for the growth of *H. zea* ranged from 10.43–21.42 °C; the precipitation seasonality suitable for the growth of *H. zea* ranged from 0–89.27; the precipitation of the driest quarter suitable for the growth of *H. zea* ranged from 34.24–431.58 mm; and the precipitation of the warmest quarter suitable for the growth of *H. zea* ranged from 93.08–620.26 mm.

### 3.3. Potential Suitable Habitats of H. zea under the Current Climate

The PGDs for *H. zea* under current climate conditions are presented in Figure 6. Our results showed that the high suitability habitat area was 3.26 × 10^4^ km^2^, accounting for 0.34% of the Chinese mainland area, and located mainly in western Zhejiang Province, eastern and southern Jiangxi Province, southeastern Hunan Province and northwestern Fujian Province; the moderate suitability habitat area was 93.45 × 10^4^ km^2^, accounting for 9.74% of the Chinese mainland area, and located mostly in Henan, Jiangsu, Anhui, Hubei, Zhejiang, Jiangxi, Hunan, Fujian, northeastern Guangxi, southern Guangdong, and southern Shanxi Provinces, as well as in sporadic areas of southern Shanxi Province, the junction of Gansu, Qinghai and Sichuan Provinces. The moderately and high suitability habitats of *H. zea* were patchily and sporadically distributed in southeastern China, respectively. Low suitability habitat was widely distributed in all provinces of China.

### 3.4. Potential Suitable Habitats, Changes, and Centroid Distributional Shifts of H. zea under Projected Climate Change

The PGDs and changes of *H. zea* under projected climate change are presented in Figure 7 and Figure 8. Our results showed that different types of suitable habitats changed greatly among the current climate, the project climate of the 2030s and the project climate of the 2050s under SSP1-2.6, SSP2-4.5 and SSP5-8.5. The shifts from unsuitable to the low suitability habitat and from the moderately to the high suitability habitat were more significant than the other shifts. The low suitability habitat mainly expanded in Yunnan, Guangxi, Sichuan and Guangdong Provinces; the high suitability habitat mainly expanded in Hunan, Hubei, Jiangxi, Fujian, Zhejiang and Anhui Provinces (Appendix A and Figure 8).

Our results showed that during the 2030s, under SSP1-2.6, the high suitability habitat area of *H. zea* was predicted to be 5.97 × 10^4^ km^2^, the moderate suitability habitat area was predicted to be 88.01 × 10^4^ km^2^, and the total habitat area was predicted to be 395.28 × 10^4^ km^2^, accounting for 0.62%, 9.17% and 41.18% of the Chinese mainland area, respectively; during the 2050s, under SSP1-2.6, the high suitability habitat area of *H. zea* was predicted to be 25.39 × 10^4^ km^2^, the moderate suitability habitat area was predicted to be 91.76 × 10^4^ km^2^, and the total habitat area was predicted to be 463.74 × 10^4^ km^2^, accounting for 2.64%, 9.56% and 48.31% of the Chinese mainland area, respectively (Appendix A). From the current to the 2030s and from the 2030s to the 2050s, under SSP1-2.6, the high suitability habitats showed a gradual increase. During the 2050s, the unsuitable habitats in South China were partially converted to the low suitability habitats, and the centroid of total suitable habitats moved southward. From the current to the 2030s, the area that the unsuitable habitat shifted to the low suitability habitat was 7.31 × 10^4^ km^2^, and the area that the moderate suitability habitat shifted to the high suitability habitat was 3.20 × 10^4^ km^2^. From the 2030s to the 2050s, the area that the unsuitable habitat shifted to the low suitability habitat was 73.64 × 10^4^ km^2^, and the area that the moderate suitability habitat shifted to the high suitability habitat was 19.32 × 10^4^ km^2^.

During the 2030s, under SSP2-4.5, the high suitability habitat area of *H. zea* was predicted to be 4.58 × 10^4^ km^2^, the moderate suitability habitat area was predicted to be 89.38 × 10^4^ km^2^, and the total habitat area was predicted to be 397.44 × 10^4^ km^2^, accounting for 0.48%, 9.31% and 41.4% of the Chinese mainland area, respectively; during the 2050s, under SSP2-4.5, the high suitability habitat area of *H. zea* was predicted to be 18.19 × 10^4^ km^2^, the moderate suitability habitat area was predicted to be 63.5 × 10^4^ km^2^, and the total habitat area was predicted to be 380.45 × 10^4^ km^2^, accounting for 1.89%, 6.61% and 39.63% of the Chinese mainland area, respectively (Appendix A). From the present day to the 2030s to the 2050s, under SSP2-4.5, the conversion of moderate suitability habitat to high suitability habitat showed a significant increase, while poorly and moderate suitability habitat did not vary much. From the present day to the 2030s, the area that the moderate suitability habitat shifted to the high suitability habitat was 2.07 × 10^4^ km^2^; from the 2030s to the 2050s, the area that the moderate suitability habitat shifted to the high suitability habitat was 13.79 × 10^4^ km^2^.

During the 2030s, under SSP5-8.5, the high suitability habitat area of *H. zea* was predicted to be 5.43 × 10^4^ km^2^, the moderate suitability habitat area was predicted to be 88.4 × 10^4^ km^2^, and the total habitat area was predicted to be 388.93 × 10^4^ km^2^, accounting for 0.57%, 9.21% and 40.51% of the Chinese mainland area, respectively; during the 2050s, under SSP5-8.5, the high suitability habitat area of *H. zea* was predicted to be 16.88 × 10^4^ km^2^, the moderate suitability habitat area was predicted to be 89.91 × 10^4^ km^2^, and the total habitat area was predicted to be 425.13 × 10^4^ km^2^, accounting for 1.76%, 9.37% and 44.28% of the Chinese mainland area, respectively (Appendix A). From the current to the 2030s to the 2050s, under SSP5-8.5, the trends of suitable habitat conversion were similar to those under SSP1-2.6. From the present day to the 2030s, the area that the unsuitable habitat shifted to the low suitability habitat was 14.63 × 10^4^ km^2^, and the area that the moderate suitability habitat shifted to the high suitability habitat was 2.16 × 10^4^ km^2^; from the 2030s to the 2050s, the area that the unsuitable habitat shifted to the low suitability habitat was 46.86 × 10^4^ km^2^, and the area that the moderate suitability habitat shifted to the high suitability habitat was 11.54 × 10^4^ km^2^.

In summary, during the 2030s and 2050s, under SSP1-2.6, SSP2-4.5 and SSP5-8.5, the ranges of PGDs of *H. zea* were quite different from those under the current climate, mainly in terms of the expansion of highly and low suitability habitats in Southern China. Especially during the 2050s, under SSP1-2.6, the expansion of poorly and high suitability habitats in Southern China was more significant. The area increases of poorly and high suitability habitats under SSP1-2.6 were much larger than those under SSP 5-8.5; the smallest increase was observed under SSP 2-4.5.

The centroid of the suitable habitats of *H. zea* is shown in Figure 9. Under the current climate, the centroid of suitable habitats was located at the point (108.86° E, 33.42° N). Under SSP1-2.6, the centroid of suitable habitats shifted to the point (109.50° E, 33.51° N) from the present day to the 2030s and to the point (108.87° E, 33.20° N) for the 2050s. It shifted 0.64° E and 0.09° N from the present day to the 2030s and 0.63° E and 0.31° N from the 2030s to the 2050s. Under SSP2-4.5, from the present day to the 2030s, the centroid of suitable habitats shifted to the point (109.59° E, 33.40° N) and to the point (109.58° E, 33.57° N) for the 2050s. It shifted 0.73° E and 0.02° N from the present day to the 2030s and shifted 0.72° E and 0.15° N from the 2030s to the 2050s; under SSP5-8.5, from the present day to the 2030s, the centroid of suitable habitats shifted to the point (110.40° E, 33.48° N), and it shifted to the point (109.01° E, 32.77° N) for the 2050s. It shifted 1.54° E and 0.06° N from the present day to the 2030s and shifted 0.15° E and 0.65° N from the 2030s to the 2050s. From the present day to the 2030s, under SSP1-2.6, SSP2-4.5 and SSP5-8.5, the centroid of the suitable habitats of *H. zea* showed a general tendency to move eastward; from 2030s to the 2050s, under SSP1-2.6 and SSP5-8.5, it moved southward, and it moved slightly northward under SSP2-4.5. Under SSP5-8.5, the centroid moved the longest distance; under SSP1-2.6, it moved the second longest distance; and under SSP2-4.5, it moved the shortest distance.

## 4. Discussion

Based on steadily increasing global temperatures and altered precipitation, the prediction of invasive alien species’ (IAS) Potential Geographical Distributions (PGDs) can largely assist strategic and tactical decisions in IAS early warning management [28,44,45]. To date, related studies have been performed on many IAS, e.g., invasive forest insects (Mountain Pine Beetle, *Lymantria dispar*) [46,47], and invasive agriculture insects (*Phenacoccus solenopsis*) [48]. However, little information is available on the impacts of warming temperatures and changing precipitation on *H. zea* in China. *Helicoverpa* are major agricultural pests worldwide, and their occurrence range is expanding [49]. *Helicoverpa* feed on more than 200 kinds of host plants in more than 30 families [50]. *Helicoverpa zea*, a seasonal migratory insect, is an important pest of economic significance in North America, and mainly damages crops, such as cotton and maize [33]. If *H. zea* invades China, it will likely cause considerable losses to Chinese agricultural production. Therefore, in the present study, data from 1781 valid occurrence records of *H. zea* and eight environmental variables were employed to build a MaxEnt model. Thereafter, the dynamic characteristics and influencing variables of the PGDs of *H. zea* in China under climate change were clarified, and the possibility of its dispersal in China was assessed. The low suitability habitats of *H. zea* identified by the model overlap with the main cotton and maize producing areas in China, such as Xinjiang and three northeastern provinces, and the high suitability habitats identified by the model overlap with the main fruit and vegetable production areas in China. Potential expansion of poorly and high suitability habitats for *H. zea* could therefore present a greater risk to cotton, corn, fruit, and vegetable production in China under projected future climates. The results of the present study are not only valuable for early warning regarding *H. zea,* but also provide unique information on how IAS react to climate change, which is one of the highest research priority areas and an integral part of the practical programs and projects of the 2030 agenda and sustainable development goals of the Food and Agriculture Organization (FAO) of the United Nations [51].

In recent years, China has become one of the countries that is most seriously affected by biological invasions, which have greatly damaged natural environments and agricultural production [52]. China has identified 560 IAS, 92 of which damage agroecosystems [53]. Adults of *H. zea* can be carried hundreds of kilometers by wind, and the larvae can be transported across oceans with their hosts [54]. China has a total of 253 international ports of entry [55]. In September 2014, inspection and quarantine at Gongbei customs (Guangdong Province) intercepted a batch of insects in a container imported from the United States; these insects were identified to include the first imported *H. zea* intercepted at Chinese ports. This event showed that *H. zea* can be accidentally transported in goods, packing containers, and cargo containers.

Insects are very sensitive to the features of the external environment, such as temperature and precipitation or humidity. Therefore, climate change will certainly have an impact on the survival and development of insects [56]. The direct impacts of climate change on insects are mainly reflected in the increase in temperature affecting the growth and development (overwintering survival rate), metabolic rate, number of generations, survival, reproduction and other life activities of insects [57,58]. Precipitation also changes under climate change, which has direct and indirect effects on insects. Some smaller insects are affected by heavy rainfall, which can influence their population size. Precipitation also changes the relative humidity of the air, which affects host plants and indirectly affects insects [59]. Our results showed that the environmental variables significantly affecting the PGDs of *H. zea* were the mean diurnal range (bio2), precipitation seasonality (bio15), precipitation of the driest quarter (bio17) and precipitation of the warmest quarter (bio18). Butler cultured *H. zea* on maize at different temperatures and reported total larval development times of 31.8, 28.9, 22.4, 15.3, 13.6 and 12.6 days at 20.0, 22.5, 25.0, 30.0, 32.0 and 34.0 °C, respectively [60]. These results showed that the developmental duration of *H. zea* became significantly shorter with increasing temperatures and that the development of *H. zea* responded significantly to temperature changes. As the climate warms, precipitation and temperature in the coastal areas of southeastern China will continue to increase [61]. Moreover, the mean annual temperature and mean diurnal range of suitable habitats for *H. zea* in China were >10 °C, and the mean annual temperature and mean diurnal range of high suitability habitat were >15 °C. Temperatures above these thresholds may be more favorable for the survival of the host plants of *H. zea* and itself, thereby increasing its survival probability. Overall, the temperature conditions under the current and future climates in southern China are suitable for the survival of *H. zea*; thus, this region faces a risk of colonization and dispersal of *H. zea*. All of the above findings further verified the accuracy of our results that temperature and precipitation have significant effects on the survival of *H. zea*. In our study, we selected the bioclimatic variables of the Beijing Climate Center Climate System Model-Middle Resolution (BCC-CSM2-MR) to predict the PGDs of *H. zea* under climate change. This is because, for different atmospheric model resolutions, the BCC-CSM2-MR optimized the parameters of uncertainty sensitive parameters in the physical process to set different parameter values, which makes the model able to more reasonably reproduce the characteristics of climate distribution [62]. Therefore, we predicted the PGDs of *H. zea* under climate change based on the BCC-CSM2-MR.

Predicting the suitable habitats of *H. zea* is an important part of the risk analysis of *H. zea* invasion. Early warning regarding *H. zea* can be obtained by predicting its potential suitable habitats or dispersal direction and is of great practical significance in guiding relevant departments or personnel to implement scientific prevention and control strategies in the potential area of invasion. Our results showed that some expansion of the suitable habitats of *H. zea* occurred and that the centroid of the suitable habitats shifted. Yan et al. investigated the potential global distributional shifts of poikilothermic invasive crop pest species under climate change [63], and the results of an ecological niche modeling analysis of 76 species suggested that climate change may expand the overall global distributions of pest species. Tang et al. used the MaxEnt model to predict the impact of climate change on pine wilt disease in China. The result showed that the suitable habitats for pine wilt disease also increased under climate change [26]. The geographic distributions of many IAS have changed, and such species have become more invasive due to climate change and frequent trade between countries around the world [64,65]; our results suggested that *H. zea* also becomes more invasive and has an increasing geographic range under climate change. Due to climate warming, insects are spreading at an average rate of 6.1 km per decade [66]. Because higher temperatures improve their overwintering survival rate, unsuitable habitats for insects are transformed into habitats suitable for their survival [67]. Lopez-Vaamonde et al. reported that many Lepidoptera species established populations and expanded their ranges in Europe [68]. Parmesan et al. studied nonmigratory European butterflies and showed that the geographic ranges of 63% had shifted northward and 3% had shifted southward in the 20th century [69]. An increase in temperature significantly increased the survival rate and population size of *H. armigera*, and significantly increased the larval abundance of Lepidoptera [70,71]. Our results indicated that, based on an increase in the temperature in Southeast China, the PGDs of *H. zea* will increase. Meanwhile, climate change increases the expansion risk of *H. zea* in China. From the viewpoint of climatic conditions and dispersal mode, *H. zea* has a high risk of colonization and poses an invasion risk in China. Early warning measures should be developed accordingly. Customs should implement strict quarantine measures for host plants and containers associated with *H. zea* and strengthen information exchange and cooperation, as well as conduct early-warning surveillance to reduce the risk of *H. zea* invasion.

## 5. Conclusions

The MaxEnt model based on optimized parameters for predicting suitable habitats of IAS can yield more accurate results than other approaches. We used 1781 valid distribution records of *H. zea*, eight environmental variables and a calibrated MaxEnt model to predict the suitable habitats of *H. zea* under climate change. The results showed that the model fit was excellent and that the significant environmental variables affecting the potential suitable habitats were mean diurnal range, precipitation seasonality, precipitation of the driest quarter and precipitation of the warmest quarter. The moderately and the high suitability habitats of *H. zea* were patchily and sporadically distributed in southeastern China, respectively. The low suitability habitat was widely distributed in all provinces of China under the current climate. During the 2030s and 2050s, under SSP1-2.6, SSP2-4.5 and SSP5-8.5, the PGDs for *H. zea* will expand. Especially during the 2050s, under SSP1-2.6, the expansion of the poorly and the high suitability habitats will become obvious. The conversion of unsuitable habitat to the poorly and the moderately to the high suitability habitat is significant under climate change. From the present day to the 2030s, under SSP1-2.6, SSP2-4.5 and SSP5-8.5, the centroid of the suitable habitats of *H. zea* showed a general tendency to move eastward; from 2030s to the 2050s, under SSP1-2.6 and SSP5-8.5, it moved southward, and it moved slightly northward under SSP2-4.5. *Helicoverpa zea* poses a high risk of biological invasion in China.

## Figures and Tables

**Figure 1 insects-13-00079-f001:**
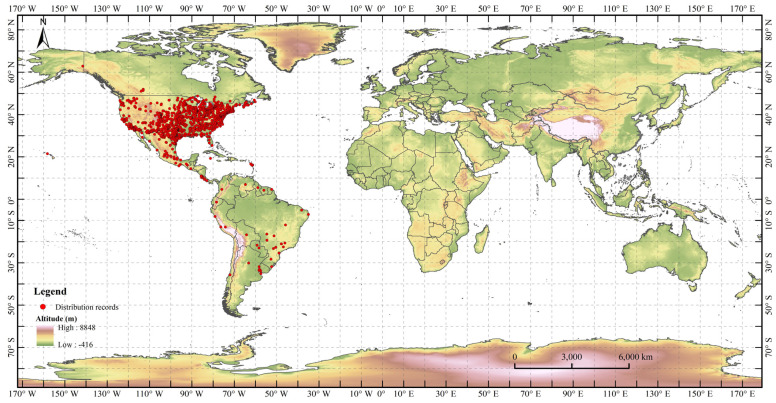
Distribution records of *Helicoverpa zea* included in MaxEnt model.

**Figure 2 insects-13-00079-f002:**
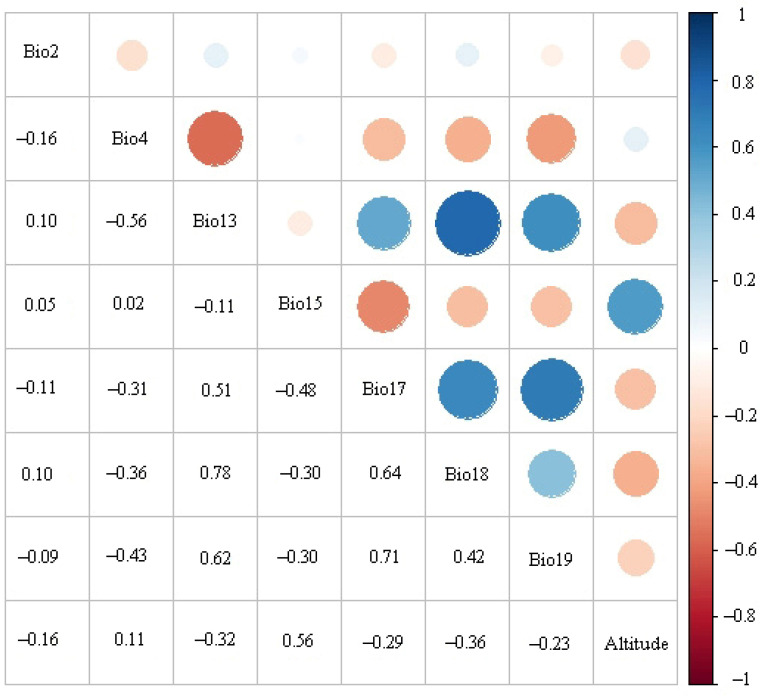
Pearson correlation coefficients for the eight environmental variables retained for MaxEnt modeling.

**Figure 3 insects-13-00079-f003:**
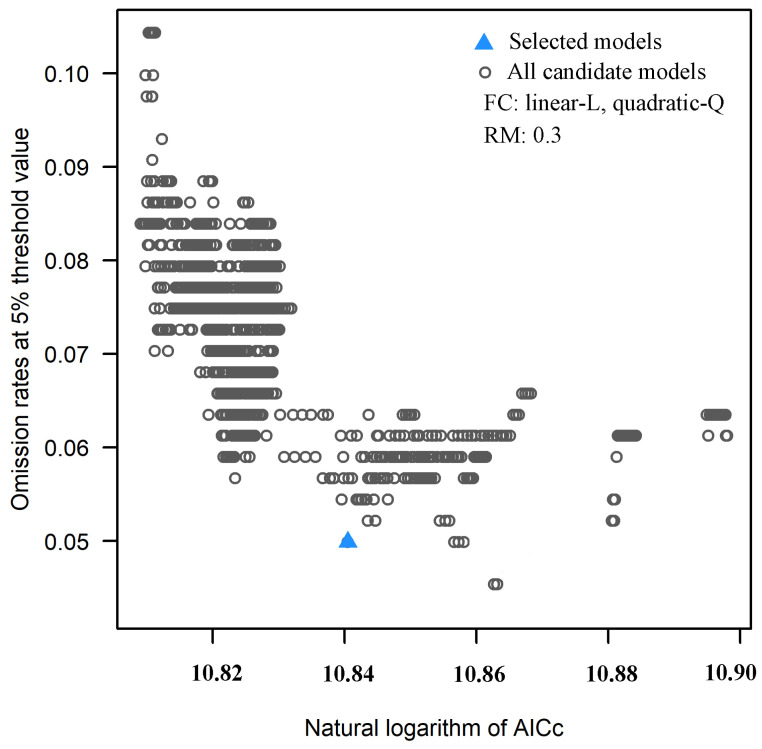
Omission rates and AICc values for all, nonsignificant, and selected ‘‘best’’ candidate models for *Helicoverpa zea*.

**Figure 4 insects-13-00079-f004:**
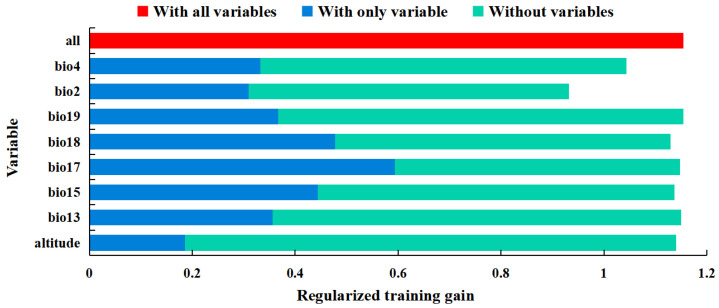
The jackknife method results of environmental variables for *Helicoverpa zea*.

**Figure 5 insects-13-00079-f005:**
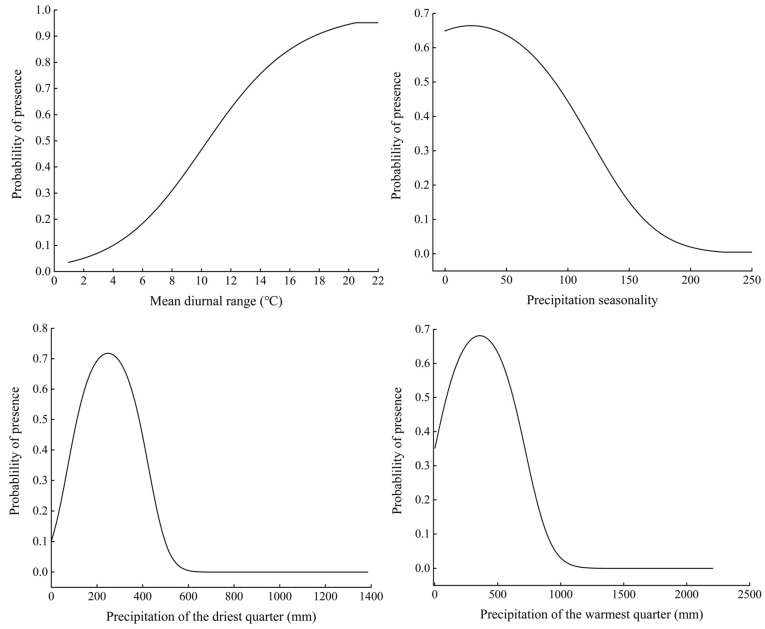
Response curves of probability of presence for *Helicoverpa zea*.

**Figure 6 insects-13-00079-f006:**
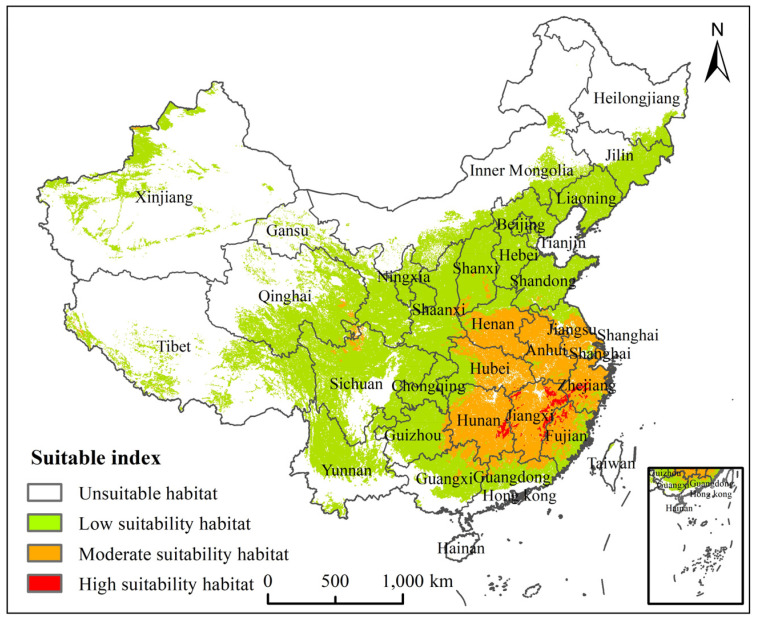
Potential suitable habitats for *Helicoverpa zea* in China under current climate.

**Figure 7 insects-13-00079-f007:**
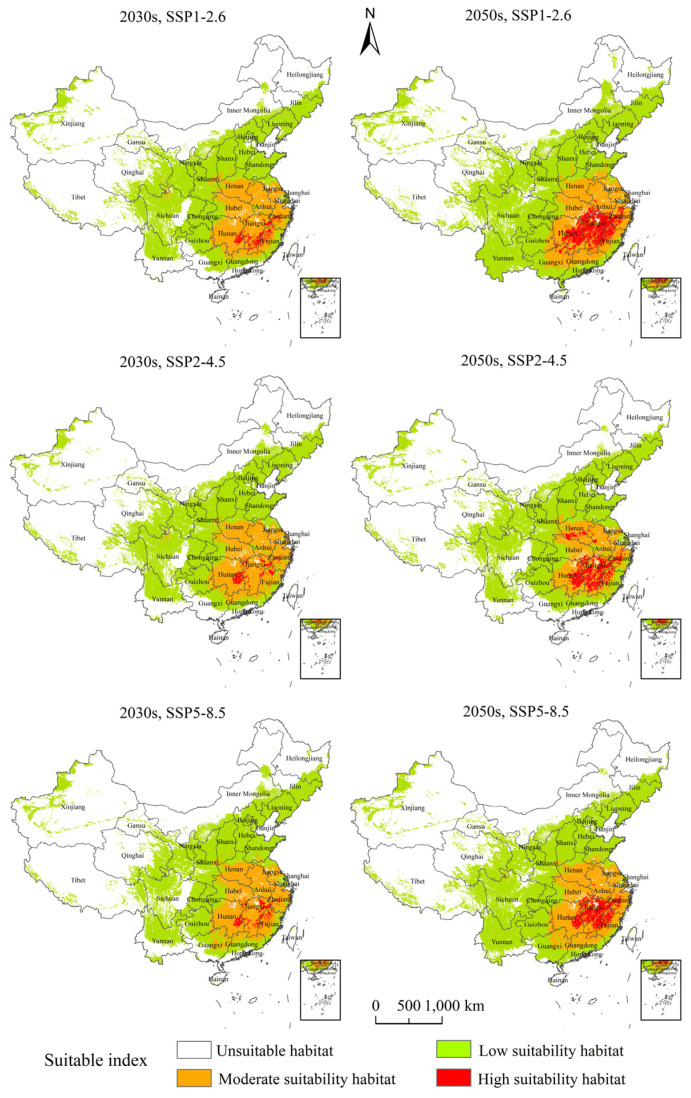
Potential suitable habitats of *Helicoverpa zea* under different climate change scenarios during the 2030s and 2050s in China. Note: SSP: shared socioeconomic pathways (see Table 2).

**Figure 8 insects-13-00079-f008:**
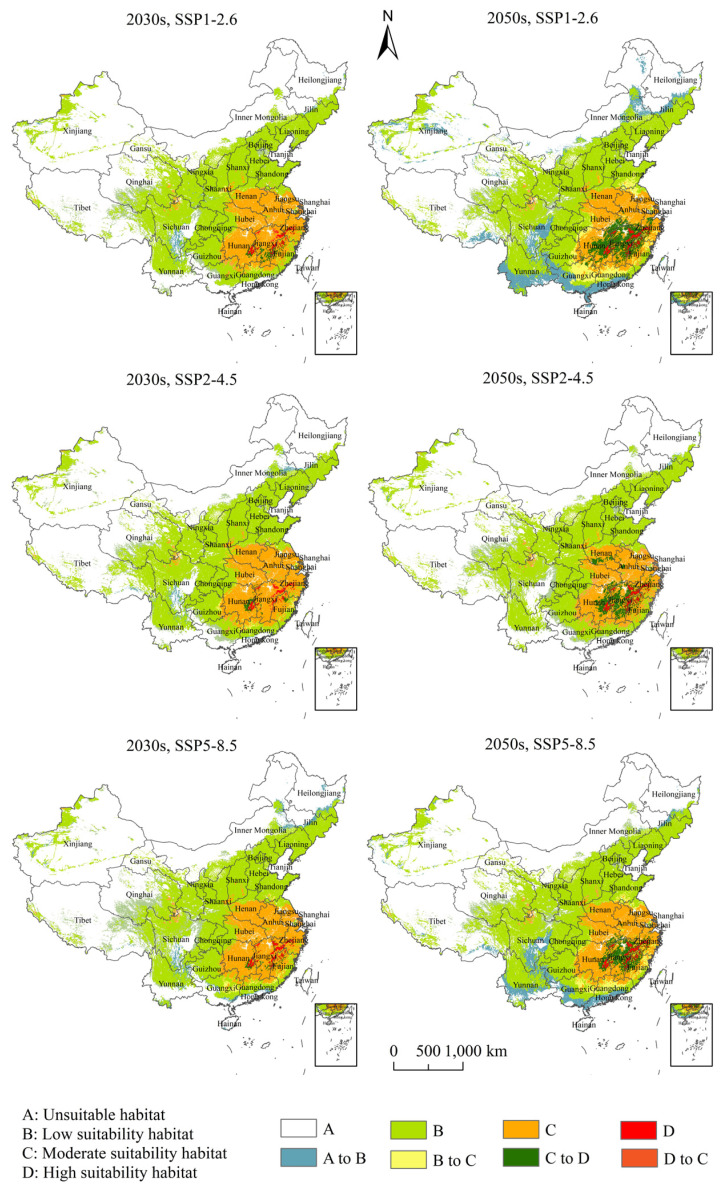
Changes in potential suitable habitats of *Helicoverpa zea* under different climate change scenarios from current to 2030s and from 2030s to 2050s in China. Note: SSP: shared socioeconomic pathways (see Table 2).

**Figure 9 insects-13-00079-f009:**
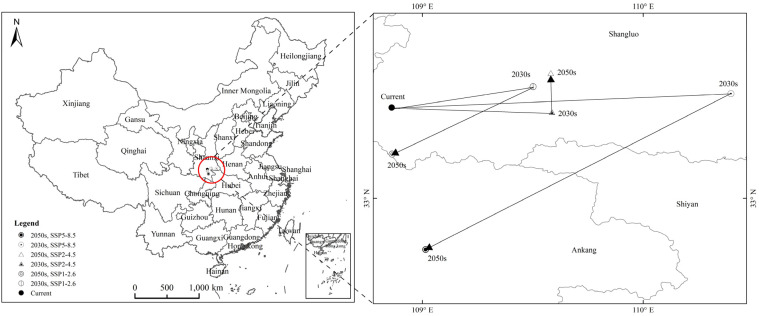
Changes in the centroid distributional shifts of *Helicoverpa zea* under climate change. Red circle: the centroid of potential geographical distribution of *H. zea*.

**Table 1 insects-13-00079-t001:** Environmental variables related to the distribution of *Helicoverpa zea*.

Variable	Description	In the Model (Yes/No)	Unit
Bio1	Annual mean temperature	No	°C
Bio2	Mean diurnal range	Yes	°C
Bio3	Isothermality	No	-
Bio4	Temperature seasonality	Yes	°C
Bio5	Max temperature of the warmest month	No	°C
Bio6	Min temperature of the coldest month	No	°C
Bio7	Temperature annual range	No	°C
Bio8	Mean temperature of the wettest quarter	No	°C
Bio9	Mean temperature of the driest quarter	No	°C
Bio10	Mean temperature of the warmest quarter	No	°C
Bio11	Mean temperature of the coldest quarter	No	°C
Bio12	Annual precipitation	No	mm
Bio13	Precipitation of the wettest month	Yes	mm
Bio14	Precipitation of the driest month	No	mm
Bio15	Precipitation seasonality	Yes	-
Bio16	Precipitation of the wettest quarter	No	mm
Bio17	Precipitation of the driest quarter	Yes	mm
Bio18	Precipitation of the warmest quarter	Yes	mm
Bio19	Precipitation of the coldest quarter	Yes	mm
Altitude	Altitude	Yes	m

**Table 2 insects-13-00079-t002:** Explanation of the three shared socioeconomic pathways.

Pathways	Description
SSP1-2.6	A world of sustainability-focused growth and equality, radiative forcing stabilizes at 2.6 W/m^2^ in 2100
SSP2-4.5	A “middle of the road” world where trends broadly follow their historical patterns, radiative forcing stabilizes at 4.5 W/m^2^ in 2100
SSP5-8.5	A world of rapid and unconstrained growth in economic output and energy use, radiative forcing stabilizes at 8.5 W/m^2^ in 2100

## Data Availability

The data presented in this study are available in this article.

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
