# Peer review of "Climate Change Increases the Expansion Risk of *Helicoverpa zea* in China According to Potential Geographical Distribution Estimation"

_insects, 2022, doi:10.3390/insects13010079_

Round 1
Reviewer 1 Report
The paper is interesting because it address an important worlwide pest, Heliothis zea. In general, the paper present the analysis of the current and future potential distribution of the referred pest in an appropriate way. However, there are some points that need to be addressed to improve and clarify the results and conclusions. First, it need to check the language style and clarify some sentences. The paper refers in several sections the global warming and relates this phenomenon as a factor in the results presented, however, no analysis of the increase in temperatures was performed so it needs to make this clear. Also, the use of a single GCM narrow the inference and scope of the results, and need to be remarked in the discussion, other GCM may yield different results. MaxEnt is not a simulation model, however, in the manuscript is considered as such and need to be corrected. Finally, the use of the elevation layer is not recommended because it is usually highly correlation with the other variables. It is remarkable that this variable is not included in the correlation analysis presented in the supplement data. If elevation is considered, it would probably be correlated with several variables and may be excluded from the analysis. Other comments are included in the manuscript.

Author Response
Thank you for your letter and for the reviewers’ comments concerning our manuscript entitled “Climate changes increase the expansion risk of Helicoverpa zea in China according to potential geographical distribution estimation” (ID:insects-1482223). Those comments are all valuable and very helpful for revising and improving our paper, as well as the important guiding significance to our researches. We have studied comments carefully and have made correction which we hope meet with approval. Revised parts are marked up using the “Track Changes” function in the paper. The main corrections in the paper and the responds to the reviewer’s commentsare as the attachment. Please see the attachment.

Reviewer 2 Report
Using existing distribution data, statistical models, and current and predicted future climate information, the authors analyzed the potential distribution of an American pest, Helicoverpa zea, in PRC, if introduced. There have been a lot of research in this field. Methods of the present study are not novel, but findings are new. Information provided is useful for prediction and control of this invasive pest in China in the future. However, the manuscript is generally wordy and sometimes repetitive. There are many awkward expressions and grammatic errors. It needs proofreading and corrections by a native English speaker in this field. I offer some specific comments below.
Abstract
Lines 21, 37 and 40: Avoid using abbreviations such as SSP1-2.6, SSP2-4.5 and SSP5-8.5 in Abstract
Introduction
Lines 49-66: Wordy, repetitive, and poorly written.
Line 56-58: this is not a sentence.
Lines 60-62: This does not make sense.
Lines 71-74: Logical and grammatical problems.
Lines 74-77: Logical links need improving.
Lines 78-93: Wordy. Use concise language.
Lines 78-80: No point to say ‘Helicoverpa zea belongs to the genus Helicoverpa’
Lines 78-84: You need references to support these statements.
Lines 87-89: This sentence needs rewriting, e.g., this is not a pest of EPPO.
Lines 91-92: This is not a sentence.
Lines 94-106: Make your aims and objectives clearer and stronger.
Lines 94-96: This sentence needs rewriting.
Lines 96-99: Incorrect sentence grammatically and logically.
Lines: 99-102: Make these points clearer.
M&M
Lines 127: Give more information on or better define ‘three shared socioeconomic pathways (SSP5-8.5, SSP2-4.5, and SSP1-2.6)’.
Line 133: What about the elevation variable?
Lines 155-158: Any references support this operation?
Results
This is unnecessarily lengthy. Please just report your key findings using concise language. Some sections need combining.
Lines 182: Make these clearer as I cannot see them straightaway in Fig. 2.
Lines 192-197: These sentences/arguments should be Introduction and /or Discussion. You should not make any argument or discussion in Results section. This section should simply describe your KEY findings.
Lines: 198-200: I am not sure if you should list these here. You’d better include these in Discussion when you explain your findings or delete them altogether.
Line 217: Why -22.7? Is the unit mm?
Line 228: ‘highly’ should read ‘moderately’.
Lines 240-263: Very wordy. Use concise language and avoid repetitions.
Line 267: ‘the expansion of poorly and highly suitable habitats is more obvious’ is vague. Any statistical data to support this statement?
Lines 273-315: Very lengthy and wordy.
Discussion
You used 3 lengthy paragraphs to address your findings, which are somewhat short in clear explanation without sharp focuses.
Lines 317-337: Wordy. Some statements need references to support.
Lines 319-320: Grammatically and logically confusing.
Lines 321-322: This needs references to support.
Conclusion is generally good.
Author Response

(The authors gave the same response as above.)

Reviewer 3 Report
The authors present a work using known species distributions in China and then compare it to current climate and future climate predictions to discuss habitat expansion. The authors do a nice job of outlining the problem and describing how they are going to analyze the data to discuss range expansion. I had a few issues with organization in the results but the main issue is that the discussion doesn’t really discuss their results but more reiterates the introduction (see specific comments).
Specific comments:
Line 192: Not entirely sure what you mean here? That they were not chosen consistently or that the variables were different among studies?
Figure 3. We need some sort of key for what the bio variables are.
Figure 5 legend. Under current, what? Incomplete sentence.
Lines 239-263: Maybe remind the reader what SSPI-2.6, SSP2-4.5, etc are so that there is some context when comparing these paragraphs.
Figure 6. On the figure too, making a notation of what SSPI1-2.6 etc are will make the reader be able to quickly understand the implications of the models and how they differ.
Section 3.5: Content here could easily be worked into section 3.4 which may make that easier to read as a whole. I like how section 3.5 gives more of a description of the changes. Combining that with the numerical results in section 3.4 would make the story seem more complete.
Figure 7: I think that the dark green coloring on C to D and D to B is confusing or at least it’s not intuitive.
Lines 304-314: Instead of just listing the centroids and their locations, maybe say it shifted 3 degrees E or N or whatever and then give the reference. When you just list the locations, it might be hard for the reader to constantly have to calculate how much a centroid is shifting.
Discussion: I feel like the results are never actually discussed in the discussion section. I feel like the rationale for doing the study is discussed at length and there is information about climate and the ability of H. zea to invade, but it never really ties back to the results with concrete examples. I suggest you work in the big results from your work in the first paragraphs in order to really point to how your work shows the range expansion possibilities and then talk about what that could mean for agricultural crops.
Author Response

(The authors gave the same response as above.)

Round 2
Reviewer 1 Report
Manuscript has been extensively rewritten and modified according to previous suggestions, minor error still prevail, marked in yellow within the manuscript. Adding province labels to the maps would really help to link information in the text to distribution values in the maps. Remember the journal has a worldwide audience who may not know Chinese province locations. Also, risk potential of damage to crops in the future requieres the assumption that current crop production areas remain the same in future years.

Author Response
Thank you for your letter and for the reviewers’ comments concerning our manuscript entitled “Climate changes increase the expansion risk of Helicoverpa zea in China according to potential geographical distribution estimation” (ID:insects-1482223). Those comments are all valuable and very helpful for revising and improving our paper, as well as the important guiding significance to our researches. We have studied comments carefully and have made correction which we hope meet with approval. Revised parts are marked up using the “Track Changes” function in the paper. The main corrections in the paper and the responds to the reviewer’s comments are as flowing:
Point 1: Adding province labels to the maps would really help to link information in the text to distribution values in the maps. Remember the journal has a worldwide audience who may not know Chinese province locations. Also, risk potential of damage to crops in the future requieres the assumption that current crop production areas remain the same in future years.
Response 1: According to your opinion, we had added province labels to the maps.
Point 2: Four or eight? You may refer that four out of the eight variables contributed the most to the model performance.
Response 2: According to your opinion, we had rewritten this sentence.
Point 3: Put a mark on selected variables, no boldface.
Response 3: According to your opinion, we had put a mark on selected variables, and other minor error had been modified in this paper.
Finally, all modified words and sentences had marked up using the “Track Changes” function in the paper.
Special thanks to you for your good comments.
We tried our best to improve the manuscript and made some changes in the manuscript. These changes will not influence the content and framework of the paper. And here we did not list the changes but marked in red in revised paper.
We appreciate for Editors/Reviewers’ warm work earnestly, and hope that the correction will meet with approval.
Once again, thank you very much for your comments and suggestions.
